# TEMPORAL-AWARE TEST-TIME TRAINING VIA SELF-DISTILLATION FOR ONE-SHOT IMAGE-TO-VIDEO SEGMENTATION

## ABSTRACT

This paper introduces a novel task and approach for one-shot medical video object segmentation using static image datasets. We address the critical challenge of limited annotated video data in medical imaging by proposing a framework that leverages readily available labeled static images to segment objects in medical videos with minimal annotation—specifically, a ground truth mask for only the first frame. Our method comprises training a one-shot segmentation model exclusively on images, followed by adapting it to medical videos through a test-time training strategy. This strategy incorporates a memory mechanism to utilize spatiotemporal context and employs self-distillation to maintain generalization capabilities. To facilitate research in this domain, we present OS-I2V-Seg, a comprehensive dataset comprising 28 categories in images and 4 categories in videos, totaling 68,416 image/frame-mask pairs. Extensive experiments demonstrate the efficacy of our approach in this extremely low-data regime for video object segmentation, establishing baseline performance on OS-I2V-Seg. The code and data will be made publicly available.

## 1 INTRODUCTION

Medical image segmentation plays a crucial role in various clinical applications, including diagnosis, treatment planning, and surgical guidance. In recent years, few-shot segmentation has gained significant attention due to its ability to learn and segment new classes with minimal annotated examples. This approach is particularly valuable in medical imaging, where obtaining large-scale, annotated datasets can be challenging and resource-intensive.

Few-shot semantic image segmentation has been extensively studied in computer vision, yielding a plethora of innovative methods. These approaches can be broadly categorized into two main paradigms: prototype learning (Snell et al., 2017; Li et al., 2021; Wang et al., 2024a) and matching-based approaches (Vinyals et al., 2016; Lu et al., 2021; Peng et al., 2023). Notably, prototypical networks have gained substantial traction in the medical domain, demonstrating particular efficacy in tasks such as organ and lesion segmentation (Roy et al., 2020; Li et al., 2023).

From image to video, recent works (Chen et al., 2021; Yan et al., 2023) explore segmenting objects of novel categories in query videos using only a few annotated support frames. Nevertheless, these approaches still necessitate densely annotated frames for training, which does not fully address the fundamental issue of annotation costs. This limitation is particularly pronounced in the medical field, where videos are not routinely recorded or stored, making them scarce. The relative paucity of publicly available medical video datasets poses significant hurdles for developing robust medical video object segmentation algorithms.

In light of these constraints, we propose a novel task: one-shot medical video object segmentation using static image datasets. Our goal is to segment a medical video with minimal annotation—specifically, a ground truth mask for the first frame—along with a corpus of labeled static images. This task setting represents an extremely low-data regime for video object segmentation, minimizing the need for costly pixel-level video annotations while maximizing the utility of more readily available image data. To achieve this, we propose a straightforward yet powerful framework. Our approach begins with training a one-shot segmentation model using labeled images. We then freeze

this model and adapt a copy of it by incorporating our proposed memory mechanism, enabling the utilization of spatiotemporal context in the target domain of medical videos. In addition, we introduce a self-distillation method to ensure that the adapted model does not deviate significantly from the original image model. This preserves generalization capabilities acquired from the diverse image dataset while allowing for domain-specific adaptations.

To facilitate research in one-shot image-to-video segmentation in medical imaging, we compile OS-I2V-Seg, a comprehensive collection of open-access medical image and video datasets with diverse classes. It includes 28 categories in images and 4 categories in videos, comprising a total of 68,416 image/frame-mask pairs.

Our contributions are summarized as follows:

- We introduce one-shot image-to-video segmentation, addressing the critical need for efficient segmentation methods in low-data video regimes.
- To tackle this challenge, we propose a test-time training method that incorporates a memory mechanism and self-distilled regularization.
- We conduct thorough experiments to validate our approach and establish baseline performance on OS-I2V-Seg.

## 2 RELATED WORK

**Few-Shot Semantic Segmentation.** Few-shot segmentation addresses the scarcity of pixel-wise annotations by enabling the segmentation of unseen classes with limited labeled data. Dong & Xing (2018) pioneer this approach by generating a single prototype vector per class from support images for comparison with query features. Subsequent studies extend this concept (Zhang et al., 2019; Liu et al., 2020; Du et al., 2022) and adapt it for medical image segmentation (Roy et al., 2020; Feng et al., 2021; Quan et al., 2022). Some approaches (Lu et al., 2021; Min et al., 2021) explore learning dense correspondences between query and support images. To address poor performance due to significant appearance changes, Peng et al. (2023) introduce a hierarchically decoupled matching network. Recently, Zhu et al. (2024) leverage large language models for improved results. However, the majority of existing few-shot semantic segmentation approaches assume that base and novel classes are sampled from the same domain. This assumption may lead to diminished performance when training and test data originate from different domains, a scenario particularly relevant to our work where we seek to generalize models trained on static images to videos.

**Cross-Domain Few-Shot Segmentation.** Cross-domain few-shot segmentation tackles a more realistic scenario where test examples differ in data distribution and label space from training data. Lei et al. (2022) introduce a feature transformation layer to map query and support features into a domain-agnostic space. Huang et al. (2023) focus on preserving intra-domain knowledge, while Su et al. (2024) propose learning rectification parameters for effective domain adaptation. We compare our approach with these cross-domain few-shot segmentation methods in Section 4.2.

**Domain Generalization.** Domain generalization aims to generalize models to diverse target domains without access to target domain data during training. Data augmentation strategies, including image transformations, feature enhancement, and generative approaches (Zhou et al., 2021; Xu & Zhao, 2023; Zheng et al., 2024), play a crucial role. Meta learning and adversarial learning (Chen & Shuai, 2021; Choi et al., 2021; Gokhale et al., 2023; Wang et al., 2024b) are also employed to tackle the challenge of domain generalization. Nevertheless, domain generalization methods encounter significant challenges when confronted with few-shot scenarios in novel domains. The scarcity of labeled samples and the presence of previously unseen categories severely test the models' capacities for effective generalization.

**Test-Time Training.** Test-time training or adaptation focuses on updating a trained model using unlabeled test samples to enhance its robustness to distribution shifts. Schneider et al. (2020) propose replacing trained statistics of normalization layers with test sample estimates. Sun et al. (2020) develop a Y-shape model with an auxiliary head for fine-tuning during inference by predicting rotation degrees. Wang et al. (2021) adapt normalization layers through test entropy minimization.

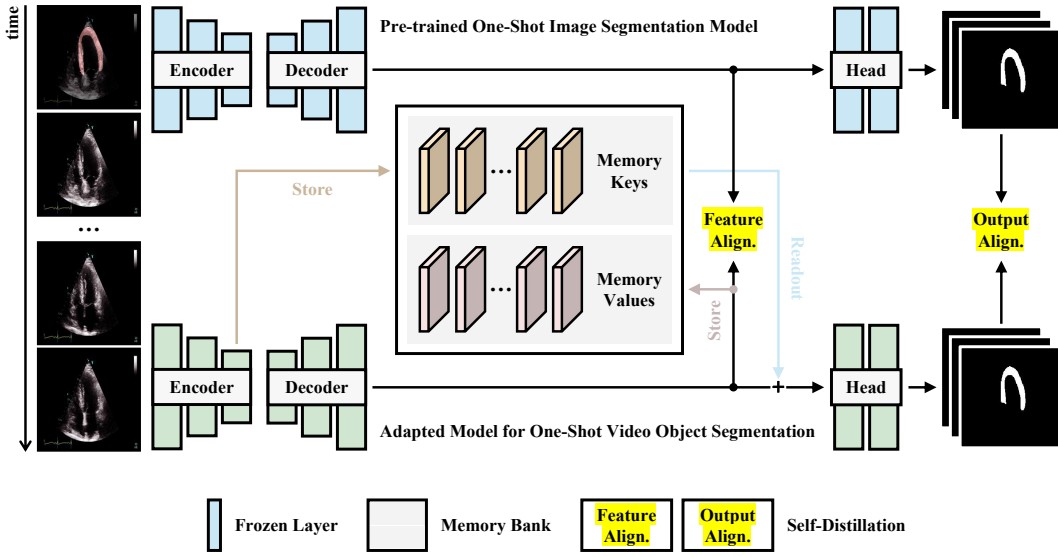

Figure 1: Our proposed temporal-aware test-time training framework for one-shot image-to-video segmentation in medical imaging. We employ a duplicated architecture of a pre-trained one-shot image segmentation model and incorporate a memory mechanism to exploit spatiotemporal context for medical video segmentation. The model is optimized using a self-distillation approach, ensuring that the adapted model does not deviate significantly from the original image model. This strategy preserves generalization capabilities acquired from diverse images while enabling domain-specific adaptations.

Gandelsman et al. (2022) introduce image reconstruction using masked autoencoders for model adaptation, while Gao et al. (2023) employ diffusion models to project target domain data into the source domain during testing. In this paper, we propose a temporal-aware test-time training method to perform one-shot image-to-video segmentation.

## 3 METHOD

### 3.1 PROBLEM SETTING

In the task of one-shot image-to-video segmentation, distinct datasets are utilized for training and testing. Specifically, the training dataset, denoted as $(\mathcal{X}_I, \mathcal{Y}_I)$, solely contains images, while the test dataset, $(\mathcal{X}_V, \mathcal{Y}_V)$, comprises videos. Here, $\mathcal{X}$ and $\mathcal{Y}$ represent input and label spaces across both datasets, which are non-overlapping, i.e., $\mathcal{X}_I \neq \mathcal{X}_V$ and $\mathcal{Y}_I \cap \mathcal{Y}_V = \emptyset$.

To manage the one-shot scenario, we implement the episodic paradigm (Vinyals et al., 2016), organizing both the training and test datasets into episodes. Each episode comprises a support set and a query set from a specific class, consisting of examples and their corresponding ground-truth labels $(\boldsymbol{x}, \boldsymbol{y}) \in \mathcal{X} \times \mathcal{Y}$. A model is trained to predict masks for multiple query images after being provided with one annotated image from the support set. During testing, the model's performance is evaluated based on its ability to segment video frames following the initially annotated frame.

### 3.2 LEARNING ONE-SHOT SEGMENTATION WITH IMAGES

Our approach begins with training a one-shot segmentation model exclusively on images. This model consists of four key components: a backbone network, a prior mask generation module, a multi-scale feature enhancement module, and a segmentation head.

Images from both the support and query sets are fed into a shared backbone network to extract mid- and high-level features. For this work, we utilize a ResNet-50 (He et al., 2016), pre-trained on ImageNet, as our backbone. The prior mask generation module employs a correlation mechanism

to process high-level feature representations of support and query images, conditioned on support masks. This generates prior masks indicating pixel probabilities of belonging to target classes, as discussed by (Peng et al., 2023). These prior masks, concatenated with mid-level features, are then fed into the multi-scale feature enhancement module (Jiang et al., 2022). This module enriches semantic representations of the query images and produces final features. The segmentation head, comprising a $3 \times 3$ convolutional layer followed by a $1 \times 1$ convolution with softmax, predicts binary segmentation masks based on these enhanced feature representations.

### 3.3 TEMPORAL-AWARE TEST-TIME TRAINING VIA SELF-DISTILLATION

While the aforementioned one-shot segmentation model, trained on static images, can be applied to video sequences by utilizing the annotated initial frame as support data and treating subsequent frames as individual query inputs, this approach disregards temporal correlations inherent in video content. Neglecting inter-frame dependencies may lead to inconsistent segmentations across the video. Although techniques such as adapters (Pan et al., 2022) can enable pre-trained image models to incorporate spatio-temporal reasoning capabilities, these methods typically require additional labeled videos, contradicting our goal of minimizing annotation dependence. To address this challenge, we propose a temporal-aware test-time training approach (cf. Figure 1). Our method duplicates the architecture and weights of the pre-trained one-shot image segmentation model and incorporates a memory mechanism. This mechanism carries long-term historical guidance to enhance the segmentation of the current frame. Through our proposed self-distillation, we optimize this augmented network without the need for extra labeled data.

**Memory Mechanism.** Instead of using a category-specific memory bank (Gong et al., 2022; Yu et al., 2024), we build a category-agnostic memory bank compatible with the one-shot segmentation setting. The memory bank is initialized with the first frame of a given video as the initial memory frame. Subsequently, we uniformly sample every fourth frame from the query set (i.e., the remaining frames) as additional memory frames. For each memory frame, we store two items: a memory key $\boldsymbol{k}^{\mathcal{M}(t)}$ and a memory value $\boldsymbol{v}^{\mathcal{M}(t)}$, where $t$ represents the index of memory frames within the memory bank. All memory frames, except the first, follow a first-in, first-out principle.

For each query frame, we first extract high-level features as detailed in Section 3.2, denoted as $\boldsymbol{q}$. Note that the memory key $\boldsymbol{k}^{\mathcal{M}(t)}$ is directly reused from the corresponding $\boldsymbol{q}$, without extra computation. The memory value $\boldsymbol{v}^{\mathcal{M}(t)}$ represents the enhanced feature representations of the corresponding query frame, derived before the segmentation head.

For memory reading, given a query frame and $T$ memory frames, we compute a pairwise affinity matrix $\boldsymbol{S}$ that quantifies the similarity between the query and each memory key:

$$\boldsymbol{S}_{ij}^{(t)} = -\frac{\|\boldsymbol{q}_i - \boldsymbol{k}_j^{\mathcal{M}(t)}\|^2}{\sqrt{C}} \, , \tag{1}$$

where $t = 1, 2, \ldots, T$, $\boldsymbol{q}_i$ denotes the feature vector at the $i$-th spatial position, with $i$ indexing over all spatial locations. Similarly, $\boldsymbol{k}_j^{\mathcal{M}(t)}$ represents the feature vector at the $j$-th position of the $t$-th memory key. Following standard practice (Vaswani et al., 2017), we normalize the matrix by $\sqrt{C}$, where $C$ is the channel dimension. To mitigate noise in memory values, we filter the affinities by retaining only top-$k$ entries. Subsequently, we employ softmax normalization on these top-$k$ entries to obtain a normalized affinity matrix $\boldsymbol{W}$ for the query:

$$\boldsymbol{W}_{ij}^{(t)} = \begin{cases} \frac{\exp(\boldsymbol{S}_{ij}^{(t)})}{\sum_{n \in \text{top-}k} \exp(\boldsymbol{S}_{in}^{(t)})} \, , & \text{if } \boldsymbol{S}_{ij}^{(t)} \in \text{top-}k \text{ of } \boldsymbol{S}_i^{(t)} \, , \\ 0 \, , & \text{otherwise} \, . \end{cases} \tag{2}$$

With $\boldsymbol{W}$, we compute a readout feature representation for the query frame as a weighted sum of memory values using an efficient matrix multiplication:

$$\boldsymbol{v}^{\mathcal{R}} = \boldsymbol{v}^{\mathcal{M}} \boldsymbol{W} \, . \tag{3}$$

This memory mechanism enables the integration of temporal context from memory frames into the current query frame's representation, enhancing the model's ability to leverage inter-frame dependencies in video segmentation tasks.

**Self-Distilled Regularization.** Having integrated temporal cues into the one-shot segmentation model trained exclusively on static images, we now face the challenge of optimizing this new model. We introduce a self-distilled regularization strategy. This approach involves freezing the original one-shot segmentation model to serve as a guide for the new model, ensuring that its features and outputs do not significantly deviate from those of the original network while accommodating temporal information.

For feature alignment, we encourage similar feature representations for the same frames in the latent spaces of both networks, thereby increasing the reliability of the adapted model's training. This necessitates the selection of appropriate layers for alignment. Given that features preceding the segmentation head are stored as memory values in the memory bank, we focus on aligning these features to mitigate the risk of error accumulation. Formally, we match features from the frozen original network ($\boldsymbol{f}^{\mathcal{A}}$) with those of the adapted network ($\boldsymbol{f}^{\mathcal{B}}$). To quantify the similarity between these feature distributions, we employ the Kullback-Leibler (KL) divergence, which is formulated as follows:

$$\mathcal{D}_{\mathrm{KL}}\left(\sigma(\boldsymbol{f}^{\mathcal{B}})\|\sigma(\boldsymbol{f}^{\mathcal{A}})\right) = \sum_j \sigma(\boldsymbol{f}^{\mathcal{B}})_j \log \frac{\sigma(\boldsymbol{f}^{\mathcal{B}})_j}{\sigma(\boldsymbol{f}^{\mathcal{A}})_j}, \tag{4}$$

where $\sigma(\cdot)$ denotes the softmax function applied to the feature vector at each spatial position, and $j$ indexes the feature vector at the $j$-th spatial position.

For output alignment, we minimize the discrepancy between segmentation masks predicted by the two models. For a given query frame, we concatenate $\boldsymbol{f}^{\mathcal{B}}$ and $\boldsymbol{v}^{\mathcal{R}}$ along the channel dimension to obtain a fused feature representation. This fused representation is then fed into the segmentation head of the adapted network to generate a segmentation mask, $\hat{\boldsymbol{y}}$. Concurrently, the query frame is input into the original network to produce another segmentation mask, $\widetilde{\boldsymbol{y}}$. To ensure output consistency, we employ a bootstrapped cross-entropy method:

$$\mathcal{L}_{\mathrm{o}} = \{\hat{\boldsymbol{y}} < \eta\}\mathbf{H}(\widetilde{\boldsymbol{y}}, \hat{\boldsymbol{y}}), \tag{5}$$

where $\mathbf{H}(\cdot)$ denotes the cross-entropy loss. To prevent overfitting on easily samples, we calculate the loss only for pixels with probabilities below a threshold $\eta$.

The overall objective function can be formulated as:

$$\mathcal{L} = \alpha\mathcal{L}_{\mathrm{o}} - \lambda\mathcal{D}_{\mathrm{KL}}\left(\sigma(\boldsymbol{f}^{\mathcal{A}})\|\sigma(\boldsymbol{f}^{\mathcal{B}})\right), \tag{6}$$

where $\alpha$ and $\lambda$ are coefficients that balance the two loss terms.

Through this self-distillation process, we optimize the memory-augmented model for one-shot video object segmentation while preserving the prior well-trained segmentation capability.

## 4 EXPERIMENTS

### 4.1 EXPERIMENTAL SETUP

**Datasets.** For training the one-shot image segmentation model, we compile a diverse set of medical image datasets: the Breast Ultrasound Images (BUSI) dataset (Al-Dhabyani et al., 2020), the TN3K dataset (Gong et al., 2023), the Multi-Modality Ovarian Tumor Ultrasound (MMOTU) dataset (Zhao et al., 2022), the Laryngeal Endoscopic dataset (Laves et al., 2019), the Brain Tumor Segmentation dataset[1], the QaTa-COV19 dataset[2], the COVID-19 CT Scan Lesion Segmentation dataset[3], the Digital Retinal Images for Vessel Extraction (DRIVE) dataset (Staal et al., 2004), the Structured Analysis of the Retina (STARE) dataset (Hoover et al., 2000), the CHASE_DB1 dataset (Fraz et al., 2012), and the ISIC Challenge datasets[4],[5]. These datasets encompass various organs and lesions, including breast nodules, thyroid nodules, ovarian tumors, larynx, brain

---

[1]https://www.kaggle.com/datasets/nikhilroxtomar/brain-tumor-segmentation

[2]https://www.kaggle.com/datasets/aysendegerli/qatacov19-dataset

[3]https://www.kaggle.com/datasets/maedemaftouni/covid19-ct-scan-lesion-segmentation-dataset

[4]https://challenge.isic-archive.com/data/#2017

[5]https://challenge.isic-archive.com/data/#2018

|  | HMC-QU | | ASU-Mayo | | CAMUS | | | | | |
|  |  | | | | LV$_{ENDO}$ | | LV$_{EPI}$ | | LA | |
|  | Dice | IoU | Dice | IoU | Dice | IoU | Dice | IoU | Dice | IoU |
|---|---|---|---|---|---|---|---|---|---|---|
| One-Shot Segmentation | | | | | | | | | | |
| PANet | 60.98 | 44.54 | 49.72 | 37.56 | 54.19 | 39.68 | 55.81 | 40.43 | 56.00 | 41.39 |
| HSNet | 70.44 | 55.17 | 58.53 | 46.05 | 63.16 | 49.52 | 63.85 | 50.22 | 65.34 | 51.20 |
| DCAMA | 67.44 | 51.58 | 53.27 | 42.86 | 60.80 | 46.14 | 61.63 | 46.85 | 62.08 | 47.71 |
| VAT | 68.05 | 54.46 | 51.29 | 39.15 | 61.37 | 48.43 | 62.15 | 49.18 | 62.61 | 50.22 |
| SSP | 72.38 | 58.13 | 54.02 | 42.24 | 64.47 | 52.19 | 65.25 | 52.50 | 67.01 | 54.03 |
| AFA | 78.86 | 65.39 | 59.17 | 46.11 | 70.10 | 58.71 | 71.07 | 59.12 | 73.08 | 60.54 |
| SCCAN | 73.81 | 59.67 | 54.36 | 42.79 | 65.33 | 52.96 | 66.31 | 53.72 | 67.49 | 55.04 |
| DCP | 78.44 | 64.71 | 57.70 | 43.48 | 70.38 | 57.91 | 70.62 | 58.31 | 71.91 | 59.98 |
| Cross-Domain One-Shot Segmentation | | | | | | | | | | |
| PATNet | 69.05 | 57.54 | 57.78 | 44.43 | 62.00 | 52.28 | 62.54 | 52.75 | 64.38 | 54.29 |
| RestNet | 74.25 | 61.87 | 55.08 | 42.40 | 67.50 | 57.01 | 67.09 | 56.65 | 68.36 | 57.68 |
| DRA | 76.07 | 61.84 | 56.42 | 43.91 | 67.84 | 54.85 | 68.67 | 55.98 | 70.60 | 57.51 |
| Test-Time Training | | | | | | | | | | |
| Tent | 81.14 | 67.02 | 60.67 | 46.78 | 71.19 | 59.04 | 71.39 | 60.16 | 73.06 | 62.42 |
| RBN | 77.01 | 64.40 | 58.58 | 44.87 | 68.18 | 56.52 | 68.76 | 58.00 | 69.97 | 59.02 |
| InTEnt | 80.41 | 67.39 | 59.22 | 45.44 | 72.52 | 60.84 | 70.06 | 59.94 | 74.05 | 63.22 |
| Ours | **82.37** | **69.02** | **61.90** | **47.45** | **73.20** | **61.86** | **74.43** | **62.36** | **76.14** | **63.77** |

Table 1: Comparative results of our method against various baselines on three public medical video datasets. Note that all test-time training methods are built upon our pre-trained one-shot image segmentation model. LV$_{ENDO}$: left ventricle endocardium; LV$_{EPI}$: left ventricle epicardium; LA: left atrium.

tumors, lung, retina, and skin lesions. To further enhance the performance of the one-shot segmentation model, we employ a training strategy that integrates natural image datasets, such as PASCAL-5i (Shaban et al., 2017), with the medical image datasets.

For evaluation, we utilize the following medical video datasets: the CAMUS dataset (Leclerc et al., 2019), the HMC-QU dataset (Degerli et al., 2021), and the ASU-Mayo Clinic Colonoscopy dataset (Tajbakhsh et al., 2016). The CAMUS dataset contains 500 patient samples, each including apical-2-chamber (A2C) and apical-4-chamber (A4C) echocardiography videos with segmentation masks for the endocardium and epicardium of left ventricle and left atrium wall. The HMC-QU dataset comprises 109 A4C view echocardiography videos with left ventricle wall segmentation masks. The ASU-Mayo dataset is a video dataset containing 38 video sequences, commonly used to build and test real-time polyp detection systems. It is important to note that our model and competing methods are trained on base image classes and evaluated on novel video classes with no class overlap, enabling assessment of generalization to unseen data.

More detailed information on these datasets is provided in Appendix B.

**Competing Methods.** In our comparative analysis, we consider three distinct methodologies. The first involves using few-shot segmentation models trained on images and directly applied to medical videos. The second is cross-domain few-shot segmentation, which extends few-shot models to generalize from source domain to other domains. The third integrates our one-shot image segmentation model with existing test-time training algorithms for one-shot image-to-video segmentation.

Note that in our experiments, all few-shot segmentation approaches are evaluated under the one-shot setting, wherein a single annotated image or frame serves as the support set.

For the first methodology, we evaluate eight representative models: PANet (Wang et al., 2019), HSNet (Min et al., 2021), DCAMA (Shi et al., 2022), VAT (Hong et al., 2022), SSP (Fan et al., 2022), AFA (Karimijafarbigloo et al., 2023), SCCAN (Xu et al., 2023), and DCP (Lang et al., 2024). Furthermore, for cross-domain few-shot segmentation, we compare our proposed approach with PATNet (Lei et al., 2022), RestNet (Huang et al., 2023), and DRA (Su et al., 2024). It is important to note that direct comparisons with test-time training algorithms specifically developed for videos are not feasible. These methods typically incorporate spatio-temporal modules (Lo et al.,

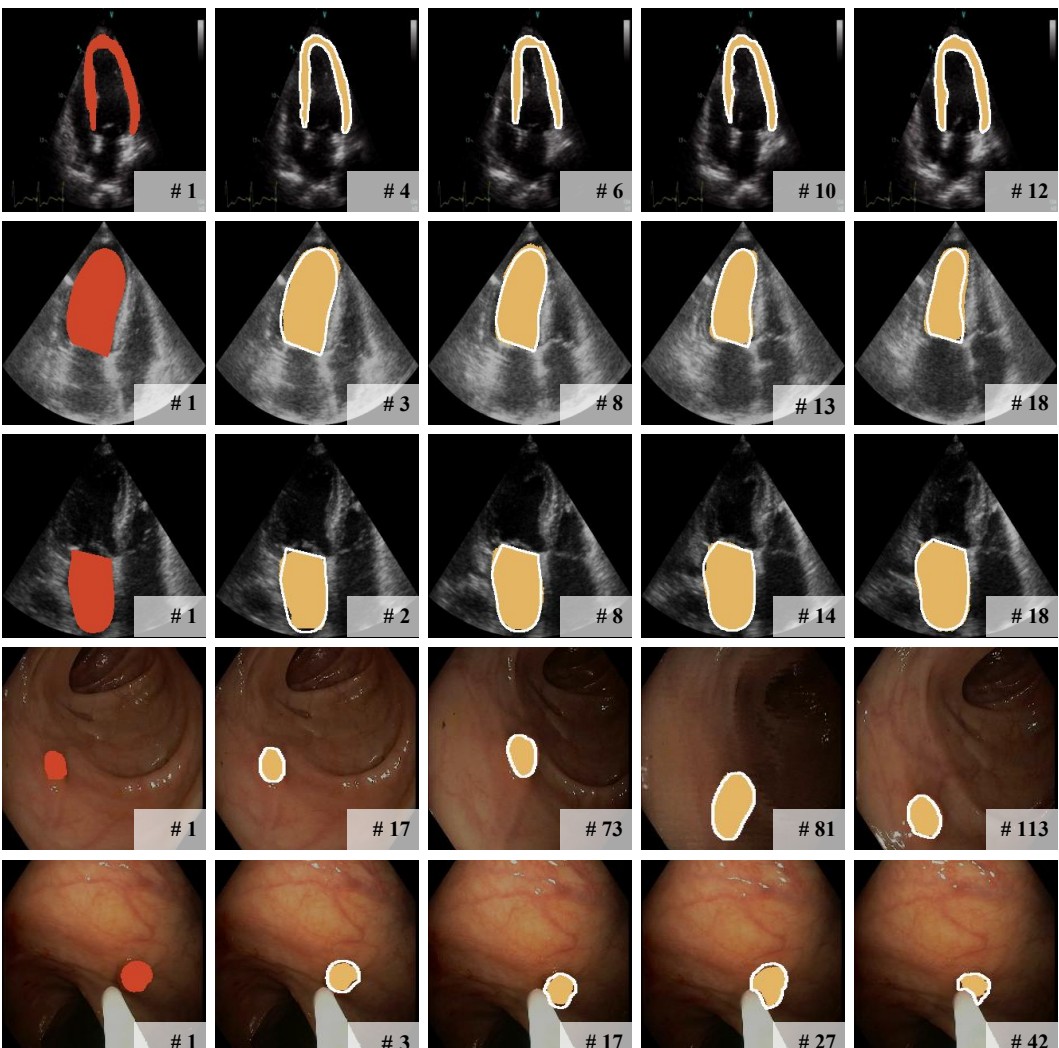

Figure 2: Qualitative results of our one-shot image-to-video segmentation model on three medical video datasets. Row 1: HMC-QU. Rows 2-3: CAMUS. Rows 4-5: ASU-Mayo. The leftmost column shows annotated support frames with ground truth masks (red). Subsequent columns showcase our model's segmentation predictions (yellow masks) on sampled query frames from videos. Ground truth masks for the query frames are delineated in white for reference.

2023; Yi et al., 2023; Su et al., 2023), whereas our base model is pre-trained on images and lacks temporal-aware components, precluding a fair comparison. Consequently, we consider the following three test-time training approaches suitable for our case: Tent (Wang et al., 2021), RBN (Benz et al., 2021), and InTEnt (Dong et al., 2024). These methods, alongside our one-shot segmentation model pre-trained on images, are applied to medical videos.

**Evaluation Metrics.** Following prior works (Ouyang et al., 2022), we adopt Dice score and foreground-background IoU as evaluation metrics to assess the performance of different models.

**Implementation Details.** To ensure a fair comparison, all models are trained on the same image datasets and evaluated on the same medical video datasets. During training of the one-shot segmentation model using images, the backbone's weights are frozen except for block #4, which remains trainable to learn more robust feature representations. The one-shot segmentation model is trained using the Adam optimizer with a learning rate of 1e-4 for the first 5K iterations, followed by the SGD optimizer with a learning rate of 1e-5 for subsequent iterations. We use a batch size of 8. Dur-

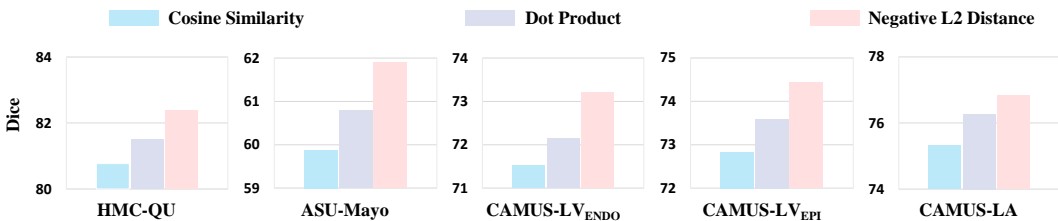

Figure 3: Ablation study on the effectiveness of different similarity measure methods.

| Mem. Bank | Feat. Align. | Pred. Align. | HMC-QU | | ASU-Mayo | | CAMUS | | | | | |
|---|---|---|---|---|---|---|---|---|---|---|---|---|
| | | | | | | | LV$_{ENDO}$ | | LV$_{EPI}$ | | LA | |
| | | | Dice | IoU | Dice | IoU | Dice | IoU | Dice | IoU | Dice | IoU |
| ✗ | ✗ | ✗ | 80.01 | 66.99 | 60.81 | 46.61 | 72.05 | 59.52 | 71.80 | 60.59 | 73.61 | 62.02 |
| ✗ | ✓ | ✓ | 77.95 | 65.48 | 58.73 | 45.66 | 69.39 | 58.30 | 70.22 | 59.10 | 71.71 | 59.93 |
| ✓ | ✗ | ✓ | 79.72 | 66.73 | 61.01 | 46.83 | 71.91 | 59.25 | 71.23 | 60.17 | 72.92 | 61.62 |
| ✓ | ✓ | ✗ | 1.27 | 1.06 | 0.85 | 0.65 | 0.64 | 0.53 | 0.93 | 0.78 | 0.81 | 0.68 |
| ✓ | ✗ | ✗ | 0.47 | 0.41 | 0.33 | 0.28 | 0.28 | 0.22 | 0.41 | 0.34 | 0.38 | 0.32 |
| ✗ | ✓ | ✗ | 0.42 | 0.39 | 0.28 | 0.22 | 0.21 | 0.17 | 0.32 | 0.26 | 0.31 | 0.26 |
| ✗ | ✗ | ✓ | 76.24 | 63.90 | 57.91 | 44.47 | 68.05 | 57.43 | 67.98 | 57.36 | 67.84 | 57.22 |
| ✓ | ✓ | ✓ | **82.37** | **69.02** | **61.90** | **47.45** | **73.20** | **61.86** | **74.43** | **62.36** | **76.14** | **63.77** |

Table 2: Ablation study on the effectiveness of components in the proposed temporal-aware test-time training approach. The first row shows results from our pre-trained one-shot image segmentation model without test-time training. Subsequent rows present results for different combinations of components in our temporal-aware test-time training strategy.

ing the test-time training stage, we use the SGD optimizer with a learning rate of 1e-5 and a batch size of 4. Our method is implemented in PyTorch and runs on NVIDIA RTX 4090 GPUs.

Regarding memory bank specifications: A single memory key-value pair occupies approximately 4 Mb. Given the typically brief duration of publicly available medical videos, we set the memory bank's length to a fixed value of 10 for efficiency, which accommodates about 2 seconds of information. As previously mentioned, we adopt a top-$k$ strategy to eliminate potential noise in memory values. Additionally, to manage the computational load of the softmax operation, we consistently select the top 20% of affinities in our experiments.

## 4.2 RESULTS

We present the performance of the competing methods in Table 1. Among one-shot segmentation models, AFA achieves the highest performance with an average Dice score of 70.45% and IoU of 57.97%. Our proposed method outperforms AFA by 3.15% in Dice and 2.91% in IoU, demonstrating its efficacy in low-data scenarios. In comparison to cross-domain one-shot segmentation models, our approach yields substantial improvements, surpassing the best results by 5.68% in average Dice score and 5.77% in IoU. These gains underscore our method's superiority not only in the one-shot setting but also in addressing cross-domain (image to video) challenges. For test-time training competitors, our model exhibits notable enhancements, exceeding the best average results by 2.11% in Dice and 1.52% in IoU. This indicates that our approach effectively leverages temporal cues to refine segmentation. Figure 2 illustrates visual examples of segmentation results generated by the proposed method for medical videos. We provide additional visual comparisons between our model and other competitors on various medical video datasets in Appendix A.

## 4.3 ABLATION STUDY

**Effect of Similarity Measure.** The similarity measure plays a crucial role in computing the affinity matrix between a query frame and memory frames. Figure 3 presents the performance of different similarity measures—cosine similarity, dot product, and negative L2 distance—across various datasets. Notably, the negative L2 distance consistently outperforms the others, achieving the highest Dice score of 82.37% and IoU of 69.02% for the HMC-QU dataset, while exhibiting lower but

competitive scores for ASU-Mayo. In contrast, the cosine similarity and dot product show moderate effectiveness, with the dot product performing slightly better overall. These findings underscore the importance of selecting an appropriate similarity measure, as performance can vary significantly based on dataset characteristics.

**Effect of Temporal-Aware Test-time Training.** To investigate the effectiveness of each component in the proposed temporal-aware test-time training strategy, we conduct comprehensive ablation studies. Table 2 reports the numerical results. We set our one-shot segmentation model pre-trained on images as the baseline, then evaluate our adapted network under different component combination settings. The optimal configuration (all components present) achieves the highest Dice and IoU scores, indicating a synergistic effect among these elements. Notably, configurations including the memory bank consistently outperform those without it, while feature alignment improves outcomes when combined with other components. Conversely, the absence of output consistency substantially diminishes performance, underscoring its importance. Overall, the results highlight the necessity of integrating all three components—memory bank, feature alignment, and prediction alignment—to maximize performance in one-shot image-to-video segmentation.

## 5 CONCLUSION

In this work, we have presented a novel approach to one-shot image-to-video segmentation for medical imaging, addressing the critical need for efficient segmentation methods in low-data video regimes. The proposed two-stage framework effectively bridges the gap between static image datasets and medical video segmentation tasks. By first training a one-shot image segmentation model and then adapting it to videos using a memory mechanism and self-distilled regularization during test-time training, we leverage strengths of both image and video data.

The introduction of OS-I2V-Seg, a comprehensive dataset spanning diverse medical imaging categories, provides a valuable resource for the research community to further explore and advance this field. Our experimental results demonstrate the efficacy of our approach in leveraging limited annotations and adapting to the spatiotemporal context of medical videos. The performance gains achieved through our method underscore the potential of utilizing readily available images to improve video segmentation tasks, particularly in the medical domain where annotated videos are scarce.

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

# APPENDIX

## A  ADDITIONAL QUALITATIVE RESULTS

For visual comparison, we select three top-performing methods representing different methodological approaches: AFA, DRA, and TENT. Figures 4, 5 and 6 show segmentation results on various datasets.

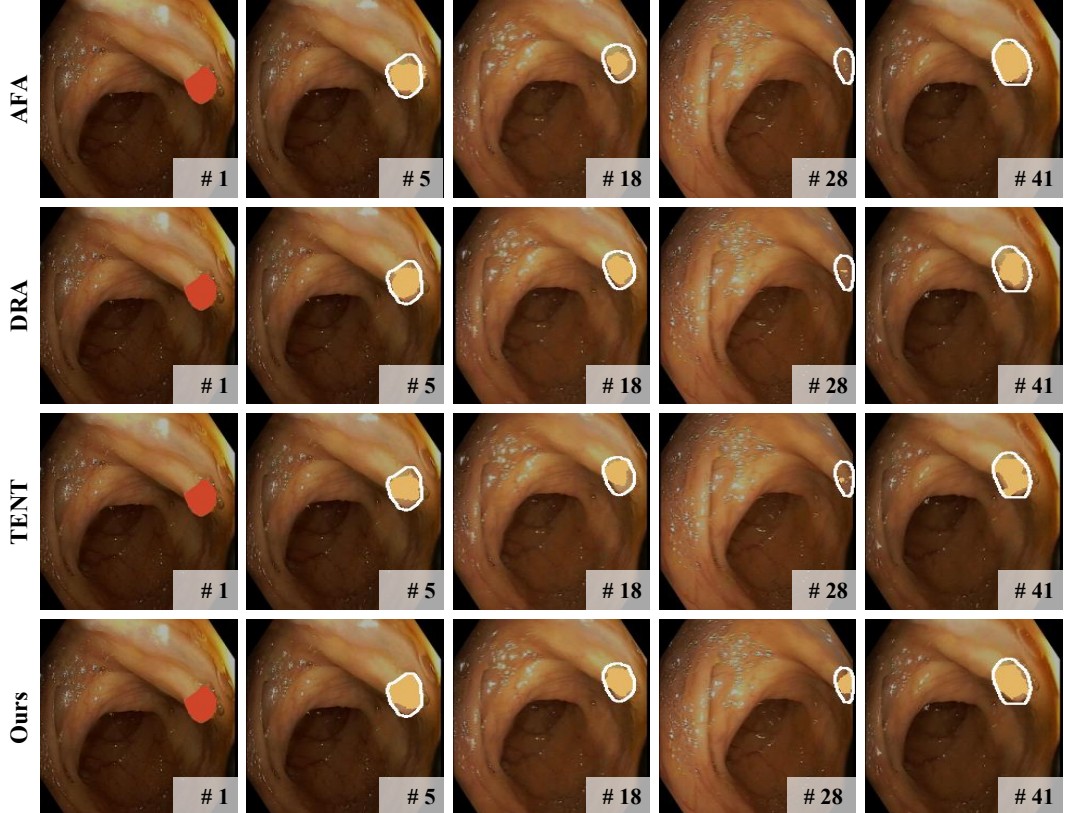

Figure 4: Qualitative results of different models on the ASU-Mayo dataset. The leftmost column shows annotated support frames with ground truth masks (red). Subsequent columns showcase segmentation predictions (yellow masks) on sampled query frames from videos. Ground truth masks for the query frames are delineated in white for reference.

# B DATASET

We visualize the distribution of organs and imaging modalities across our entire dataset in Figure 7. Below, we provide a detailed description of the training images used in our study.

**BUSI:** The dataset comprises breast ultrasound images collected in 2018 from 600 female patients aged 25 to 75. It includes 780 images, each averaging $500 \times 500$ pixels, categorized into three classes: normal, benign, and malignant. Ground truth images accompany the original images, facilitating tasks such as classification, detection, and segmentation in breast cancer diagnostics.

**TN3K:** The dataset consists of 3,493 ultrasound images collected from various imaging systems, including GE Logiq E9, ARIETTA 850, and RESONA 70B, involving 2,421 patients. Samples are selected based on the following criteria: (1) presence of at least one thyroid nodule, (2) absence of blood signals, and (3) retention of only one representative image per perspective. Each image is processed to grayscale and cropped to exclude non-ultrasound areas.

**MMOTU:** The dataset comprises 1,639 ovarian ultrasound images, including 1,469 2D images and 170 contrast-enhanced images, sourced from Beijing Shijitan Hospital, Capital Medical University. Images feature pixel-wise and global-wise annotations, captured using the Mindray Resona8 scanner. Multiple scans per patient are included, focusing on clear representations of lesion regions.

**Laryngeal Endoscopic:** The dataset contains 536 hand-segmented in vivo color images of the larynx, acquired during two different resection interventions, with a resolution of $512 \times 512$ pixels.

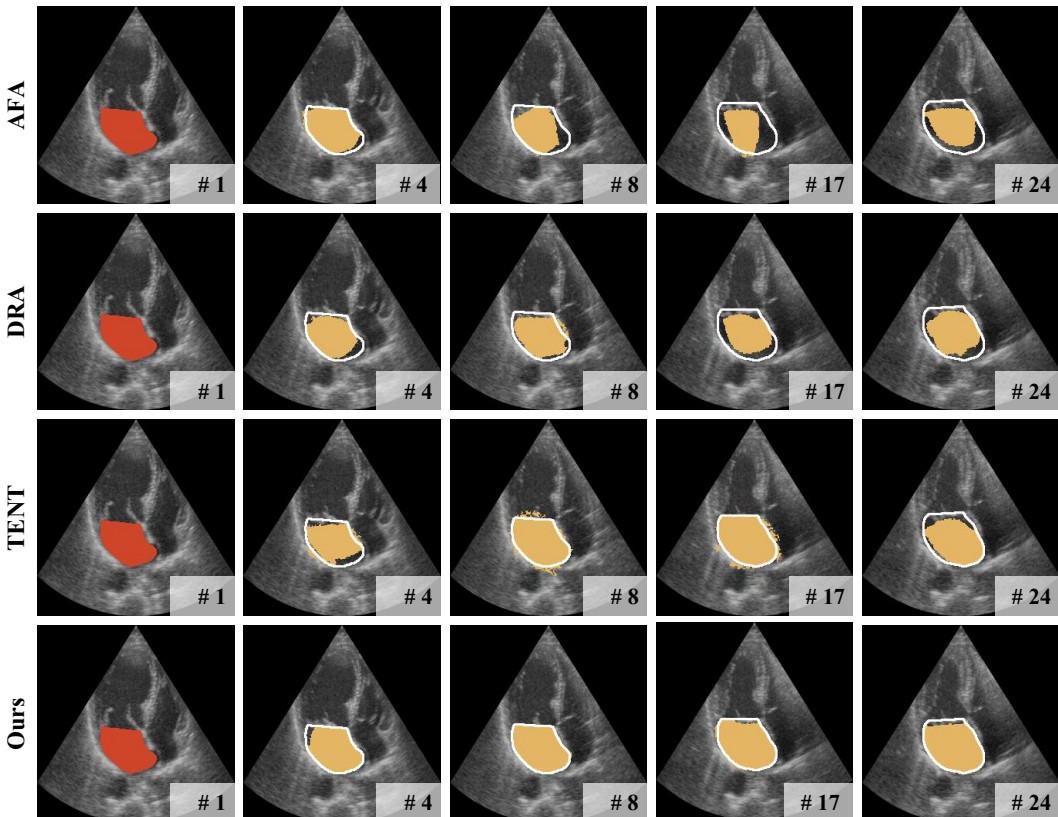

Figure 5: Qualitative results of different models on the CAMUS dataset. The leftmost column shows annotated support frames with ground truth masks (red). Subsequent columns showcase segmentation predictions (yellow masks) on sampled query frames from videos. Ground truth masks for the query frames are delineated in white for reference.

**Brain Tumor:** The dataset comprises 3,064 T1-weighted contrast-enhanced images from 233 patients, categorized into three types of brain tumors: meningioma (708 slices), glioma (1,426 slices), and pituitary tumor (930 slices).

**QaTa-COV19:** The dataset comprises 9,258 COVID-19 chest X-rays, accompanied by ground-truth segmentation masks for the COVID-19 infected regions, facilitating the segmentation task.

**COVID-19 CT Scan Lesion Segmentation:** This dataset consists of lung CT scans for COVID-19, curated from seven public datasets, including three that provide COVID-19 lesion masks. It contains 2,729 image and ground truth mask pairs, with all lesion types mapped to white for consistency.

**DRIVE:** The dataset includes 40 photographs from a diabetic retinopathy screening program in the Netherlands, featuring 400 diabetic subjects aged 25 to 90. Of the selected images, 33 show no signs of diabetic retinopathy, while 7 exhibit signs of mild early diabetic retinopathy.

**STARE:** The dataset is designed for a semantic segmentation task in medical research, comprising 397 images with pixel-level annotations for 60 labeled objects in a single class: vessels.

**CHASE_DB1:** The dataset includes 28 retinal images from both eyes of 14 children (8 white, 3 South Asian, 3 of other ethnic origins, mean age 10 years) recruited from a primary school in North-East London. Each image features two ground truth vessel segmentation annotations created by independent observers.

**ISIC Challenge:** This dataset aggregates over 20,000 publicly accessible dermoscopy images, collected from leading clinical centers worldwide and captured using various devices.

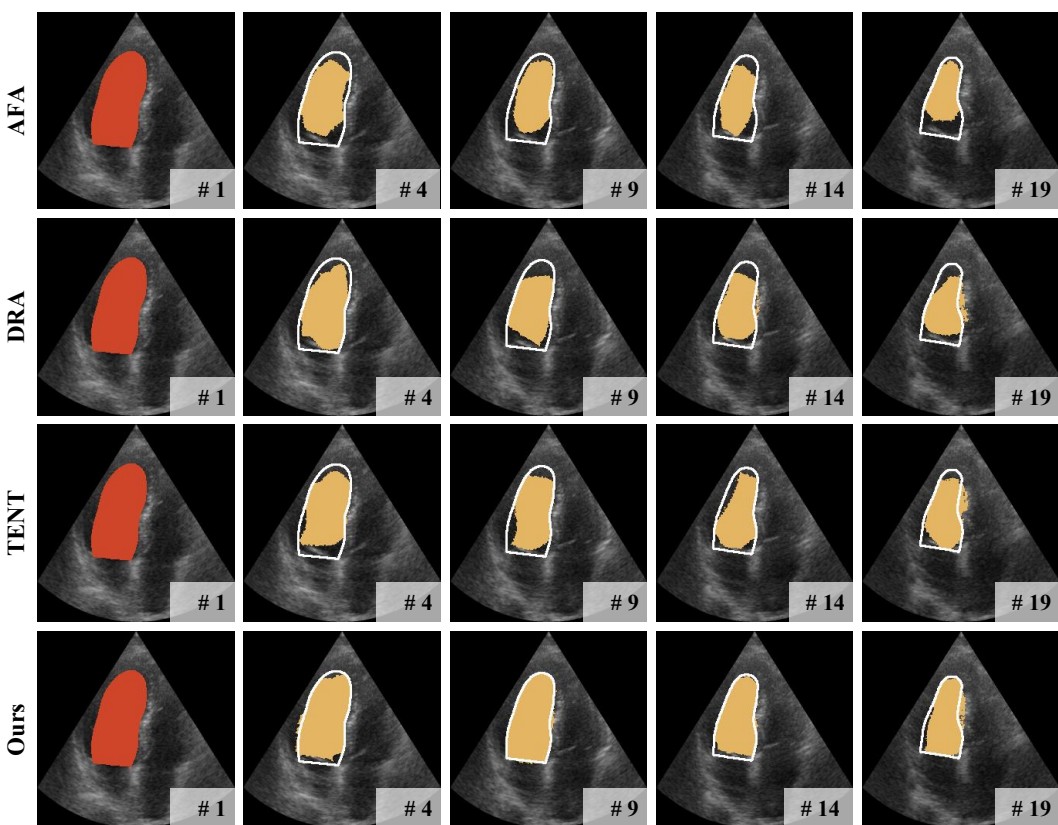

Figure 6: Qualitative results of different models on the CAMUS dataset. The leftmost column shows annotated support frames with ground truth masks (red). Subsequent columns showcase segmentation predictions (yellow masks) on sampled query frames from videos. Ground truth masks for the query frames are delineated in white for reference.

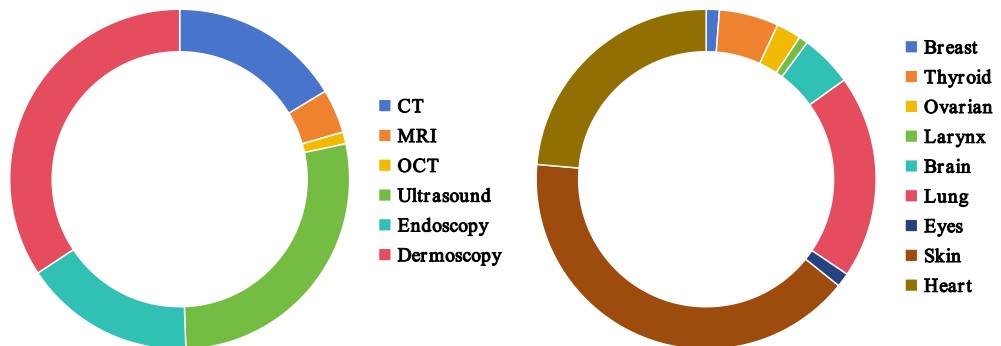

Figure 7: Imaging modality (left) and organ (right) distributions in the training set.

