# OpenReview forum: "Temporal-Aware Test-Time Training via Self-Distillation for One-Shot Image-to-Video Segmentation"
_ICLR.cc/2025/Conference — ICLR 2025 Conference Withdrawn Submission_

### Official Review · Reviewer_6Uad · 2024-10-16

**Soundness:** 3
**Presentation:** 3
**Contribution:** 2
**Rating:** 3
**Confidence:** 4

**Summary:**

This paper proposes a novel task and method for single-frame image to video segmentation, focusing on medical video object segmentation. Since annotated video data in medical images is very limited, the paper adopts a framework to achieve video object segmentation using only annotated static images, and proposes a method that uses self-distillation for training in the test phase, combined with memory mechanism to utilize spatiotemporal context, greatly reducing dependence on annotated data.

**Strengths:**

1. This paper introduces a test-time training strategy combined with a memory mechanism, enabling the model to make adaptive adjustments using spatiotemporal context during the test phase, while ensuring generalization through self-distillation.
2. The authors present a novel dataset, OS-I2V-Seg, covering 28 categories of images and 4 categories of videos, providing a valuable resource for researchers in related fields and advancing the development of medical image segmentation tasks.
3. The proposed method is not only well-suited for the task of single-frame image-to-video segmentation but also maintains strong generalization in cross-domain scenarios through the self-distillation mechanism, demonstrating excellent task scalability.

**Weaknesses:**

The research motivation of this project is somewhat lacking. For example, one of the primary issues with ECHO images or videos is the insufficient availability of high-quality annotated data, which limits the ability to train robust segmentation models. Consequently, there are several shortcomings in the current approach:

1. The proposed method requires training an image segmentation model as a prerequisite for video segmentation. However, for a novel test dataset with insufficient annotations, the image segmentation step is likely to fail, which would, in turn, lead to failure in video segmentation. Furthermore, ECHO does not require full video segmentation; only the ED and ES frames are necessary for clinical purposes. Datasets like CAMUS has ground-truth information like LVEF, authors can calculate the segmentation results of ED and ES to check the accuracy with Ground-truth LVEF.

2. To address this issue, the paper should include more ECHO video segmentation experiments to validate the model's effectiveness. While the paper primarily focuses on ECHO in its experiments, the limited number of videos in the test datasets weakens the argument. Although the CAMUS dataset contains around 500 videos, its extensive preprocessing and high quality do not represent the general clinical setting, which diminishes the persuasiveness of the results. The EchoDynamic dataset could be considered as an alternative for testing.

3. While the study aims to propose a novel image-to-video segmentation method, it is notable that widely recognized segmentation methods, such as SAM2 and BioMedSAM2, are not included in the comparison experiments for similar annotated segmentation tasks. This omission undermines the credibility of the results. The authors should explain more about why these exps not included.

4. The proposed method requires further validation, particularly in terms of time inference. Without such experiments, the current approach risks being comparable to frame-by-frame image segmentation, which would nullify the significance of this method. Time inference experiments are crucial to demonstrate the advantage of the proposed video segmentation method. It is a good idea to compare the segmentation time cost for segmenting the whole video with traditional frame-to-frame video segmentation model or SAM2.

**Questions:**

1. add exp on EchoDynamic dataset.
2. try to compare with SAM2
3. make the significance of the paper more meaningful.
4. ECHO images really requires 3-types segmentation as shown in table1. Were these three segmentation trained separately or together?

**Details Of Ethics Concerns:**

New published medical dataset needs extra privacy illustration.

---

### Official Review · Reviewer_35mV · 2024-11-03

**Soundness:** 2
**Presentation:** 1
**Contribution:** 2
**Rating:** 5
**Confidence:** 4

**Summary:**

This paper introduces a novel task and approach for one-shot medical video object segmentation using static image datasets. The authors address the challenge of limited annotated video data in medical imaging by proposing a framework that leverages readily available labeled static images to segment objects in medical videos with minimal annotation. The proposed method involves a two-stage process, training a one-shot segmentation model on images and temporal-aware test-time training via self-distillation. Experimental results demonstrate that the proposed method outperforms existing approaches in this low-data regime for video object segmentation on OS-I2V-Seg.

**Strengths:**

The paper is well-motivated. It presents a novel problem formulation by introducing one-shot medical video object segmentation using only static image datasets.

The proposed framework combines a memory mechanism with self-distillation during test-time training. This allows the model to leverage temporal information in videos without requiring additional annotated video data during training.

Extensive experiments, including comparisons with state-of-the-art methods and ablation studies, demonstrate the effectiveness of the proposed approach.

**Weaknesses:**

1. There are no details of the multi-scale feature enhancement module in the manuscript and in the figure.

2. The process of usage of memory values is not revealed in Figure 1.

3. Although ablation studies are presented, more detailed analysis on the impact of specific hyperparameters (e.g., memory bank size, selection of top-k affinities) could provide deeper insights into the method's performance and robustness.

4. The results section is not well-organized or described. It would be better to provide a more detailed description and analysis and reorganize some content from the Appendix into the manuscript within the page limitation.

5. It would be better to also collect other SOTA performance on HMC-QU, ASU-Mayo, and CAMUS to show the progress between the current work and the ultimate target based on temporal models.

6. The possible reasons behind the dramatic performance decrease are not analyzed. For example, why does no TTT generally generate sub-optimal performance, but using some modules may slightly decrease the performance, and some make training collapse, yet using them all can generate the best performance? There is no deeper analysis of the performance gap and how and why some designs work.

**Questions:**

Please see the weakness part 1,2,3,4,5,6.

---

### Official Review · Reviewer_L3oX · 2024-11-03

**Soundness:** 2
**Presentation:** 2
**Contribution:** 2
**Rating:** 3
**Confidence:** 5

**Summary:**

This paper proposes a method for one-shot video object segmentation where only the first frame of the video is annotated during testing. The main idea involves first pre-training an image segmentation backbone on the labelled static images, followed by adapting the segmentation model to medical videos through test time training strategy using a form of spatiotemporal consistency and self-distillation.

For enforcing spatiotemporal consistency, the authors proposed a FIFO memory mechanism (acting as support set) that stores the features from the segmentation backbone of support images as keys (K), and the output of segmentation head (enhanced features) as values (V). The FIFO memory always contains the first annotated frame of the video. To predict the final enhanced feature for the query frame q, authors use similarity of q’s spatial features with those of keys present in FIFO memory queue as weights for weighted average of values present in FIFO. This process can be thought of autoregressive prediction based on softmaxed similarity with past T frames observed before the current frame.

The self-distillation from the original pre-trained model helps to regularize the adapted model by preventing overfitting to the labelled frame of the video. Authors utilize Hinton’s KL divergence as teacher student distillation loss treating the softmaxed features over its dimension as the corresponding probability.

Moreover, the paper also introduces a new Dataset OS-I2V-Seg for this task.

**Strengths:**

1. The paper tackles an important problem of generalizing dense segmentation task over videos using sparsely annotated frame (in their case the first annotated frame).
2. The experiments on several medical videos show that their method outperform different one-shot, cross-domain one-shot and test-time training methods.
3. One major strength about this method is incorporation of 2D image segmentation model for 3D video frames during testing using temporal awareness though additional memory mechanism.

**Weaknesses:**

Major Weakness:
1. In general, though the main idea of the paper is conveyed clearly, the specific detail about their method is missing. For instance, details about one-shot segmentation model pretrained on the static images is missing in Section 3.2 and Figure 1. Specifically, only backbone network is shown in the figure, while no discussion on the prior mask generation module, multi-scale feature enhancement module and segmentation head is included. It’s unclear how the authors use (Peng et al. 2023)’ s correlation mechanism for prior mask generation module. If I am correct, Peng et al. 2023, use correlation between single support image’s features and the query image’s features to generate the binary mask. Do the authors use the same methodology?

2. In the introduction section, the authors mention one of the major limitations of Few Shot methods is dealing with discrepancy between the domains of base and novel classes [line 083-084]. Could the authors clarify what base class means in this context? Does it mean that the support images are from different domain compared to the query image, or it means training and testing datasets are from different domain. In either case, it is unclear how the current methodology tackles the domain shift problem. In the first scenario, there is no domain shift as the support frame is from the same video, while in the second scenario, I am not sure how the current method can overcome domain shifts between the pre-training and testing domains.

Minor Weakness:
1. In eq. 3, what are the dimension of W, v^R, v^M. It may be unclear to the readers how the read-out value is obtained exactly and further utilized and how it represents weighted average of similar features in the past frames.

2. The title of the paper should highlight medical video segmentation as the current approach is not tested for non-medical videos.

3. Size of FIFO queue. Since the performance the method is correlated to the FIFO queue, a discussion on how to set the size of queue is lacking.

**Questions:**

1. Different self-distillation regularization? The authors employ KL divergence between the features obtained from pre-trained segmentation backbone and the current adapted model. This model of self-distillation assumes that the features are “logits” for the loss computation, which might give incorrect regularization as the softmax(logits) = softmax(logits + contant). Why is this model motivated? Does this help with the domain shift problem discussed by the authors earlier?

2. Is the current method autoregressive? In other words, for predicting the segmentation mask of current query, does the method only look at the past frames? Can this be extended to bidirectional approach, where we look at all the frames for the prediction of the current query for eq 3?

3. Can the current method be extended to foundational segmentation models like SAM (Segment Anything Model)? How does current method compare to SAM? It would be good to know if the current method is a useful alternative to these foundation models based on computation, segmentation accuracy, etc.

---

### Note · Authors · 2024-11-15

I have read and agree with the venue's withdrawal policy on behalf of myself and my co-authors.